# Both IgM and IgG Antibodies against Polyethylene Glycol Can Alter the Biological Activity of Methoxy Polyethylene Glycol-Epoetin Beta in Mice

**DOI:** 10.3390/pharmaceutics12010015

**Published:** 2019-12-21

**Authors:** Tien-Ching Chang, Bing-Mae Chen, Wen-Wei Lin, Pei-Hua Yu, Yi-Wen Chiu, Yuan-Tsong Chen, Jer-Yuarn Wu, Tian-Lu Cheng, Daw-Yang Hwang, Steve Roffler

**Affiliations:** 1Institute of Biomedical Sciences, Academia Sinica, Taipei 11529, Taiwan; shelly@ibms.sinica.edu.tw (T.-C.C.);; 2Taiwan International Graduate Program in Molecular Medicine, National Yang-Ming University and Academia Sinica, Taipei 11529, Taiwan; 3Department of Laboratory Medicine, School of Medicine, College of Medicine, Kaohsiung Medical University, Kaohsiung 80708, Taiwan; 4Division of Nephrology, Kaohsiung Medical University Hospital, Kaohsiung 80708, Taiwan; 5School of Chinese Medicine, China Medical University, Taichung 40447, Taiwan; 6Department of Biomedical Science and Environmental Biology, Center for Biomarkers and Biotech Drugs, Kaohsiung Medical University, Kaohsiung 80708, Taiwan; 7Graduate Institute of Medicine, College of Medicine, Kaohsiung Medical University, Kaohsiung 80708, Taiwan; 8National Institute of Cancer Research, National Health Research Institutes, Tainan 70456, Taiwan

**Keywords:** polyethylene glycol (PEG), anti-PEG antibodies, methoxy polyethylene glycol-epoetin beta, PEG-EPO, anemia, erythropoiesis

## Abstract

Pre-existing antibodies that bind polyethylene glycol are present in about 40% of healthy individuals. It is currently unknown if pre-existing anti-polyethylene glycol (PEG) antibodies can alter the bioactivity of pegylated drugs with a single long PEG chain, which represents the majority of newly developed pegylated medicines. Methoxy polyethylene glycol-epoetin beta (PEG-EPO) contains a single 30 kDa PEG chain and is used to treat patients suffering from anemia. We find that the pre-existing human anti-PEG IgM and IgG antibodies from normal donors can bind to PEG-EPO. The prevalence and concentrations of anti-PEG IgM and IgG antibodies were also higher in patients that responded poorly to PEG-EPO. Monoclonal anti-PEG IgM and IgG antibodies at concentrations found in normal donors blocked the biological activity of PEG-EPO to stimulate the production of new erythrocytes in mice and accelerated the clearance of ^125^I-PEG-EPO, resulting in PEG-EPO accumulation primarily in the liver and spleen. Accelerated clearance by the anti-PEG IgG antibody was mediated by the Fc portion of the antibody. Importantly, infusing higher doses of PEG-EPO could compensate for the inhibitory effects of anti-PEG antibodies, suggesting that pre-existing anti-PEG antibodies can be “dosed through.” Our study indicates that the bioactivity and therapeutic activity of PEG-EPO may be reduced in patients with elevated levels of pre-existing anti-PEG antibodies. New pegylated medicines with a single long PEG chain may also be affected in patients with high levels of anti-PEG antibodies.

## 1. Introduction

Polyethylene glycol (PEG) is often attached to proteins, peptides, nucleic acids, and nanoparticles to enhance water solubility, increase resistance to proteolytic enzymes, and reduce kidney excretion. At least twenty pegylated drugs have been approved by the FDA (Table 1) and dozens of pegylated medicines are in different stages of clinical development [1]. Pegylation ranges from the random attachment of multiple short PEG chains in drugs such as Adagen (11~17 5 kDa PEG molecules) and Oncaspar (69~82 5 kDa PEG molecules) to the site-specific introduction of a single long PEG chain in drugs such as Plegridy (20 kDa PEG) and Jivi (60 kDa branched PEG) [2,3,4,5]. The attachment of a single long linear or branched PEG chain has become the standard for recent pegylated medicines [1].

PEG was long considered to be nonimmunogenic, but recent reports show that anti-PEG antibodies can be induced by injection of some types of pegylated liposomes or proteins in clinical trials [6,7,8,9,10,11,12,13,14] and animal studies [15,16,17,18,19,20,21,22,23,24]. Although anti-PEG antibodies do not appear to be induced when most pegylated human proteins are administered to patients, a large proportion of the general population never exposed to pegylated drugs has pre-existing anti-PEG antibodies in their circulation, possibly due to widespread exposure to PEG in commonly used products such as cosmetics, soaps, and medicines [10,11,12,16,25,26,27,28,29,30,31]. For example, we recently found that over 40% of 1504 normal donors had measurable anti-PEG IgM and IgG antibodies in their plasma [25].

Anti-PEG antibodies can bind and accelerate the clearance of pegylated enzymes, proteins, and nanocarriers conjugated with multiple PEG chains in animals (Table 2) [18,19,20,21,22,23,32,33,34,35,36]. High levels of induced anti-PEG antibodies can also alter the half-life and biological activity of pegylated drugs in patients [7,8,11,14]. Pre-existing anti-PEG antibodies have also been linked to allergic reactions after the infusion of pegylated medicines [29,31]. However, other studies have reported that anti-PEG IgM cannot accelerate the clearance of pegylated proteins [37]. Taken together, there is a need to investigate the effects of anti-PEG antibodies on pegylated drugs.

In contrast to many studies examining the effect of anti-PEG antibodies on pegylated drugs with multiple short PEG chains, little is known about the impact of anti-PEG antibodies on drugs modified with a single, long PEG molecule. The pegylated drug market exceeded 7 billion US dollars worldwide in 2017 and is expected to experience continued growth with the introduction of increasing numbers of biological medicines [38]. Given the high prevalence of pre-existing anti-PEG antibodies in the general population and the trend towards single-chain pegylation, there is a strong impetus to understand how pre-existing anti-PEG antibodies might affect the safety and biological activity of pegylated drugs with a single PEG modification.

Methoxy polyethylene glycol-epoetin beta (PEG-EPO) is an erythropoiesis-stimulating agent (ESA) that can activate erythropoietin receptors on erythroid progenitor cells to induce the formation of new red blood cells. PEG-EPO is therefore often used to treat patients with anemia associated with chronic kidney disease [39,40]. There are around 85,000 patients diagnosed with chronic kidney disease (0.3% of the total population) in Taiwan, and the percentage of patients requiring regular kidney dialysis is the highest in the world. Therefore, there is a high demand for PEG-EPO in Taiwan and the market of PEG-EPO worldwide is expected to reach 550 million US dollars in 2020. PEG-EPO is modified with one linear 30 kDa PEG chain to extend its half-life (approximately 130 h), thereby allowing it to be administered intravenously or subcutaneously at prolonged dosing intervals. It is currently unknown if anti-PEG antibodies can alter the bioactivity of PEG-EPO.

In the present investigation, we sought to determine if anti-PEG antibodies can bind PEG-EPO and alter its serum half-life, biodistribution, or bioactivity. We verified that pre-existing anti-PEG antibodies from normal human donors can bind to PEG-EPO. Well-defined anti-PEG IgM and IgG monoclonal antibodies developed in our lab were then used at concentrations found in normal donors to examine their effects on PEG-EPO [25]. Our results reveal that both anti-PEG IgM and IgG antibodies can profoundly alter the bioactivity, pharmacokinetics and biodistribution of PEG-EPO in mice. We also observed a higher frequency and concentration of anti-PEG antibodies in anemia patients that were classified as poor-responders to PEG-EPO. Our results suggest that pre-existing anti-PEG antibodies may alter the biological activity of PEG-EPO in some patients.

## 2. Materials and Methods

### 2.1. Reagents

NH_2_-PEG_10,000_-NH_2_ was from Iris Biotech GmbH (Marktredwitz, Germany). Methoxy polyethylene glycol-epoetin beta (PEG-EPO) was from Roche (F. Hoffmann-La Roche Ltd., Switzerland). Na^125^I was purchased from PerkinElmer Life and Analytical Sciences, Inc. (Boston, MA, USA). Hybridoma cells secreting AGP4 and 6.3 anti-PEG IgM and IgG monoclonal antibodies were previously described [32,41].

### 2.2. Ethical Statement

The studies were approved by the Institutional Review Boards and Ethics Committees of Academia Sinica (AS-IRB01-15038(F); Latest revised approval: 7 March 2016; Duration: 5 January 2016 ~ 31 December 2018) and Kaohsiung Medical University Chung-Ho Memorial Hospital (KMUHIRB-F(II)-20160117; Latest revised approval: 8 June 2018; Duration: 27 December 2016 ~ 31 August 2019) in Taiwan. Written informed consent was obtained from the subjects in accordance with institutional requirements and Declaration of Helsinki principles.

### 2.3. Plasma Sample Collection

Plasma samples from healthy Han Chinese donors residing in Taiwan were enrolled from a prior project that had been collected, centrifuged, and stored at the National Center for Genome Medicine, Academia Sinica. All donors of this study agreed to offer the remaining centrifugal plasma for research in a blinded fashion. Animal studies were approved by the Institutional Animal Care and Utilization Committee of Academia Sinica in Taiwan.

### 2.4. Human Anti-PEG Antibody Assays

To measure binding of human anti-PEG antibodies to PEG-EPO, Maxisorp 96-well microplates (Nalge-Nunc International, Rochester, NY, USA) were coated with 0.25 μg/well PEG-EPO in 50 μL/well 0.1 M NaHCO_3_/Na_2_CO_3_ (adjusted to pH 9.5 with HCl) buffer overnight at 4 °C and then blocked with 200 μL/well 5% (*w*/*v*) Difco^TM^ skim milk powder (BD, Franklin Lakes, NJ, USA) in Dulbecco’s phosphate-buffered saline (PBS) at room temperature for 2 h. Plates were washed once with PBS before serial dilutions of plasma samples containing pre-existing anti-PEG antibodies or control human antibodies (sources and identity of antibodies used in our study are listed in Appendix A) were added to plates for 1 h at room temperature. The plates were washed twice with PBS containing 0.1% 3-[(3-cholamidopropyl)dimethylammonio]-1-propanesulfonate (CHAPS) and once with PBS. Antibody binding was detected by addition of 0.25 μg mL^−1^ of secondary antibody (Appendix A) for 1 h at room temperature. The plates were washed as above before adding 100 μL/well 2,2′-azino-bis(3-ethylbenzothiazoline-6-sulphonic acid) (ABTS) containing 0.03% H_2_O_2_ for 30 min at room temperature. The absorbance (405 nm) of wells was measured in a microplate reader (Molecular Devices, San Jose, CA, USA).

### 2.5. Clinical Relevance of Anemia Patients

To investigate possible clinical relevance between anti-PEG antibody levels and therapeutic efficacy of PEG-EPO, we collected plasma samples from 55 peritoneal dialysis patients including 39 patients classified as poor responders to PEG-EPO and 16 control patients that responded well to PEG-EPO. All included patients received peritoneal dialysis with monthly subcutaneously 100 mcg PEG-EPO treatment. The hemoglobin level of the patients was recorded monthly between July 2011 to June 2016. Patients with gastrointestinal bleeding, hypoalbuminemia (<3.5 g dL^−1^), secondary hyperparathyroidism, inadequate iron storage, and inadequate dialysis ((Kurea×Td)/Vurea < 1.7) were excluded. PEG-EPO poor-responders were defined as those with hemoglobin level less than 10 g dL^−1^ who required extra erythropoietin treatment. The measurement of anti-PEG IgM and IgG prevalence and concentrations in responders (hemoglobin level > 10 g dL^−1^) and poor-responders (hemoglobin level < 10 g dL^−1^) was performed as previously described using humanized anti-PEG monoclonal antibodies cAGP4-IgM and c3.3-IgG as reference standards [25].

### 2.6. Mouse Monoclonal Anti-PEG Antibody Assay

Maxisorp 96-well microplates (Nalge-Nunc International, Rochester, NY, USA) were coated with 0.25 μg/well PEG-EPO (Roche) in 50 μL/well 0.1 M NaHCO_3_/Na_2_CO_3_ (adjusted to pH 9.5 with HCl) buffer overnight at 4 °C and then blocked with 200 μL/well 5% (*w*/*v*) skim milk powder (Difco) in Dulbecco’s phosphate-buffered saline (PBS, Thermo Fisher Scientific, Waltham, MA, USA) at room temperature for 2 h. Plates were washed once with PBS immediately before use. AGP4 IgM and 6.3 IgG were diluted from 550 ng mL^−1^ in 2% (*w*/*v*) skim milk powder with PBS and then four additional 3-fold serial dilutions were made in 2% (*w*/*v*) skim milk powder with PBS and added to plates at room temperature for 1 h. Unbound antibodies were removed by washing the plates twice with 0.1% CHAPS/PBS and once with PBS. The 0.2 μg mL^−1^ peroxidase-conjugated affinipure secondary antibodies specific to mouse IgM μ-chain and mouse IgG (Appendix A) in 50 μL PBS containing 2% (*w*/*v*) skim milk powder were added to the IgM or IgG detection plates, respectively, for 1 h at room temperature. The plates were washed as above before adding 100 μL/well ABTS containing 0.03% H_2_O_2_ for 30 min at room temperature. The absorbance (405 nm) of wells was measured in a microplate reader (Molecular Devices, San Jose, CA, USA).

### 2.7. Immunoblotting

500 ng recombinant human erythropoietin (EPO) (R&D System, Minneapolis, MN, USA) and 300 ng PEG-EPO were electrophoresed in a 10% SDS-PAGE gel under reducing conditions before overnight transfer to nitrocellulose paper by capillary diffusion in blotting buffer (50 mM NaCl, 2 mM EDTA, 0.5 mM 2-mercaptoethanol, 10 mM Tris-HCl, pH 7.5). Blots were blocked for 1 h with 5% (*w*/*v*) skim milk powder in PBS and incubated with 6.3, AGP4, or anti-human erythropoietin (Appendix A) for 1 h at RT, and then secondary antibodies with peroxidase-conjugated affinipure specific to mouse IgG and mouse IgM μ-chain (Appendix A) for 1 h at RT. After washing three times with 0.1% CHAPS/PBS, specific bands were visualized by ECL detection according to the manufacturer’s instructions (Pierce, Rockford, IL, USA), then detected by FUJIFILM LAS-3000 imaging system (Tokyo, Japan).

### 2.8. Measurement of Anti-PEG Antibodies in Mice Serum

BALB/c mice (8~12 weeks) were intravenously injected with different doses of AGP4 (75 μg·kg^−1^, 300 μg·kg^−1^, 1 mg·kg^−1^, 5 mg·kg^−1^, 10 mg·kg^−1^), 6.3 (80 μg·kg^−1^, 480μg·kg^−1^, 3.2 mg·kg^−1^, 16 mg·kg^−1^), 6.3 (200 μg·kg^−1^, Fc portion-mediated study), and 6.3 F(ab’)_2_ (200 μg·kg^−1^, Fc portion-mediated study). At 24 h after injection, serum samples were collected and the concentrations of anti-PEG antibodies were measured by direct ELISA on PEG-coated plates using AGP4, 6.3, and 6.3 F(ab’)_2_ as reference standards and peroxidase-conjugated anti-mouse IgM, IgG, or F(ab’)_2_ specific secondary antibodies (Appendix A) (bars, SD; *n* = 4).

### 2.9. Red Blood Cell Measurement

Female BALB/c mice (8~12 weeks) were intravenously injected with anti-PEG antibodies (5 mg kg^−1^ AGP4 or 16 mg kg^−1^ 6.3) 24 h before the mice were intravenously injected with 5 μg kg^−1^ PEG-EPO. Blood was collected from the tail vein at day 7 after PEG-EPO injection. Blood was mixed with 50 μL of heparin (60 U mL^−1^) to prevent coagulation. Blood samples were diluted 10,000-fold in PBS and red blood cell numbers were measured on a Attune NxT Flow Cytometer (ThermoFisher Scientific Inc., Waltham, MA, USA).

### 2.10. ^125^I- PEG-EPO Labeling

PEG-EPO (25 μg, in 0.1 M sodium phosphate buffer, pH 7.5) and 1 mCi Na^125^I (PerkinElmer Life and Analytical Sciences, Inc., Waltham, MA) were agitated for 75 min at room temperature in a Pierce™ iodination tube (Thermo Fisher Scientific, Waltham, MA, USA). NaI was added to a final concentration of 1 mM to terminate iodide oxidation. Free iodine was removed by passage through a PD-10 desalting column (GE Healthcare Life Sciences, Boston, MA) equilibrated with PBS containing 0.25% bovine serum albumin (BSA). Radioactivity was counted on a 2480 WIZARD^2^ Automatic Gamma Counter (PerkinElmer Life and Analytical Sciences, Waltham, MA). The specific activity of radiolabeled PEG-EPO ranged from 4 × 10^5^ to 5 × 10^5^ cpm μg^−1^ [42,43].

### 2.11. Biodistribution of ^125^I- PEG-EPO

Female BALB/c mice (8–12 weeks) were intravenously injected with anti-PEG antibodies (5 mg kg^−1^ AGP4 or 16 mg kg^−1^ 6.3). At 24 h after antibodies injection, mice were intravenously injected with 5 μg kg^−1 125^I-PEG-EPO. At 1 h and 6 h after ^125^I-PEG-EPO infusion, mice were sacrificed and perfused with 25 mL PBS and then blood, spleen, liver, kidney, lung, brain, bladder and part of the small intestine were collected and weighed. Radioactivity and distribution of ^125^I-PEG-EPO in organs were quantified on a 2480 WIZARD^2^ Automatic Gamma Counter (PerkinElmer Life and Analytical Sciences, Waltham, MA). Results show percentage of radioactivity of each organ to injection dose, which was divided by the weight of each organ (% ID g^−1^).

### 2.12. Pharmacokinetics of ^125^I- PEG-EPO

Female BALB/c mice (8~12 weeks) were intravenously injected with different doses of AGP4 (75 μg kg^−1^, 300 μg kg^−1^, 1 mg kg^−1^ and 5 mg kg^−1^) or 6.3 (80 μg kg^−1^, 480 μg kg^−1^, 3.2 mg kg^−1^ and 16 mg kg^−1^) to mimic pre-existing anti-PEG antibodies in the circulation. At 24 h after antibodies injection, mice were intravenously injected with ^125^I- PEG-EPO (5 μg kg^−1^). Blood samples were collected from the tail vein at 30 min, 2 h, 6 h, and 24 h. The concentration of ^125^I- PEG-EPO in plasma was quantified by measuring radioactivity and comparison with a ^125^I- PEG-EPO standard curve. For Fc-mediated assay, mice were pre-injected with 200 μg kg^−1^ of 6.3 and 6.3 F(ab’)_2_, then 1 h after antibodies infusion, ^125^I- PEG-EPO (5 μg kg^−1^) was injected. The concentration of ^125^I- PEG-EPO was measured at 30 min, 2 h, 6 h, and 25 h.

### 2.13. Preparation of 6.3 F(ab’)_2_ Antibody

6.3 IgG was incubated for 5.5 h at 37 °C with final concentrations of 1 mg mL^−1^ ficin (Sigma, St. Louis, MO, USA) and 2 mM cysteine-HCl (Thermo Fisher Scientific, Waltham, MA) in 50 mM Tris/2 mM EDTA, pH 7. The reaction was stopped by adding 1/10 vol of 100 mM N-ethylmaleimide (Sigma) [44]. The digested product was dialyzed against PBS. 6.3 F(ab’)_2_ was purified through protein A column three times. Purity was observed by SDS-PAGE electrophoresis and binding activity of 6.3 F(ab’)_2_ was measured by direct ELISA plate coated with 0.5 μg/well NH_2_-PEG10,000-NH_2_ and detected by peroxidase-conjugated goat anti-mouse F(ab’)_2_ or peroxidase-conjugated goat anti-mouse IgG Fc (Appendix A).

### 2.14. Statistical Analysis

Statistical analysis was processed with GraphPad Prism 7 software. Results are shown as the mean with plus or minus (±) standard deviation (S.D.). All results are representative data from repeated experiments. All statistical analyses of experiments were examined through t test; a *p* value of less than 0.05 was considered significant. No statistical method was used to predetermine sample sizes.

## 3. Results

### 3.1. Pre-Existing Anti-PEG Antibodies in Healthy Donors Bind to PEG-EPO

We previously measured the prevalence and concentration of anti-PEG antibodies in 1504 healthy Han Chinese donors residing in Taiwan [25]. The concentrations of anti-PEG IgM and IgG antibodies in donor serum samples were determined by comparison with humanized anti-PEG antibody standards. Over forty percent of donors had pre-existing anti-PEG IgM antibodies with concentrations ranging from 0.1 to 57.3 µg mL^−1^ or anti-PEG IgG antibodies with concentrations ranging from 0.3 to 238 µg mL−1. To examine if serum samples containing pre-existing anti-PEG antibodies from healthy donors can bind to PEG-EPO, serial dilutions of donor plasma samples that tested positive for anti-PEG antibodies were added to ELISA plates coated with PEG-EPO. Antibody binding to PEG-EPO was detected with anti-human IgM or IgG secondary antibodies. Compared with non-binding control human IgM and IgG antibodies, human serum samples with pre-existing anti-PEG IgM (Figure 1A), or IgG (Figure 1B) antibodies (Appendix A) strongly bound to PEG-EPO. It should be noted that all samples, including the negative control antibodies, were diluted to a starting concentration of 0.25 µg mL^−1^ in human serum that tested negative for the presence of anti-PEG antibodies. These results show that serum samples containing human anti-PEG IgM or IgG antibodies can bind PEG-EPO.

### 3.2. Pre-Existing Anti-PEG Antibodies in Anemia Patients Receiving PEG-EPO

To begin to investigate whether pre-existing anti-PEG antibodies can affect the therapeutic efficacy of PEG-EPO in patients, the concentration of anti-PEG IgM and IgG antibodies were measured in serum samples from 39 peritoneal dialysis patients suffering from anemia who did not respond well to PEG-EPO and 16 control patients who did respond to PEG-EPO. 30.8% of patients (12 of 39) that poorly responded to PEG-EPO tested positive for anti-PEG antibodies while 6.25% of patients (1 of 16) who responded to PEG-EPO had anti-PEG antibodies in their plasma samples (Figure 2A). Three of the poor responders had both anti-PEG IgM and IgG in their serum samples. In addition, the concentrations of anti-PEG antibodies in poor-responders had mean values of 1.2 μg mL^−1^ IgM and 1.7 μg mL^−1^ IgG whereas the concentration of anti-PEG IgM in the PEG-EPO responder was 0.25 μg mL^−1^ (Figure 2B). These results suggest that pre-existing anti-PEG antibodies may alter the therapeutic efficacy of PEG-EPO in anemia patients.

### 3.3. Anti-PEG Antibodies Can Decrease the Bioactivity of PEG-EPO in Mice

To better understand how anti-PEG antibodies might affect the therapeutic efficacy of PEG-EPO, we utilized monoclonal anti-PEG antibodies AGP4 (IgM) and 6.3 (IgG) in a mouse model. Both AGP4 anti-PEG IgM and 6.3 anti-PEG IgG antibodies bound PEG-EPO as measured by direct ELISA with similar avidity as human anti-PEG IgM and IgG antibodies from normal donors (Appendix A). Immunoblotting also indicated that AGP4 and 6.3 bound to PEG-EPO but not erythropoietin (EPO) (Appendix A). AGP4 IgM or 6.3 IgG were intravenously injected into BALB/c mice, resulting in serum concentrations of about 58 µg mL^−1^ AGP4 IgM and 93 µg mL^−1^ 6.3 IgG 24 h later when PEG-EPO was IV administered to the mice (Appendix A). Stimulation of red blood cell (RBC) formation by PEG-EPO was assessed by counting circulating RBCs seven days later. PEG-EPO significantly increased the number of circulating RBCs as compared to untreated mice. Injection of control IgM or IgG antibodies at the same dose as anti-PEG antibodies did not affect the ability of PEG-EPO to stimulate formation of new RBCs. By contrast, both AGP4 (Figure 3A) and 6.3 (Figure 3B) anti-PEG antibodies completely blocked the generation of RBCs induced by PEG-EPO. These results show that the presence of anti-PEG antibodies in the serum of mice can block the biological activity of PEG-EPO.

### 3.4. Anti-PEG Antibodies Promote PEG-EPO Accumulation in the Spleen and Liver

To investigate if anti-PEG antibodies can alter the biodistribution of PEG-EPO, we intravenously injected into mice non-binding control antibodies or anti-PEG antibodies to achieve serum concentrations of 58 µg mL^−1^ IgM or 93 µg mL^−1^ IgG after 24 h. ^125^I-PEG-EPO was then intravenously injected and groups of four mice were sacrificed after 1 or 6 h. The mice were immediately perfused with PBS to remove blood before the radioactivity of ^125^I-PEG-EPO in organs were measured on a gamma counter. The concentration of PEG-EPO in blood was significantly reduced in mice pre-treated with AGP4 (Figure 4A,B) or 6.3 (Figure 4C,D) at both 1 h and 6 h after PEG-EPO administration. PEG-EPO accumulation in the liver and spleen significantly increased at both 1 and 6 h in mice treated with AGP4 or 6.3. By contrast, PEG-EPO in the kidneys was reduced in the presence of AGP4 or 6.3, likely reflecting the reduced amount of PEG-EPO in the blood of mice with anti-PEG antibodies. The control IgM and IgG antibodies at the same dose as the anti-PEG antibodies did not affect the biodistribution of PEG-EPO, demonstrating that altered PEG-EPO biodistribution depended on the ability of AGP4 and 6.3 to bind PEG.

### 3.5. Anti-PEG Antibodies can Accelerate the Clearance of PEG-EPO in Mice

The effect of pre-existing anti-PEG antibodies on PEG-EPO clearance from blood was examined by injecting different doses of AGP4 IgM or 6.3 IgG intravenously into BALB/c mice 24 h before intravenous injection of ^125^I-PEG-EPO. The injected doses of AGP4 and 6.3 were selected to cover the range of anti-PEG IgM and IgG antibody concentrations measured in healthy donors [25]. Measurement of anti-PEG antibodies in serum after 24 h showed that the concentration of AGP4 ranged from 0.6 to 57.8 µg mL^−1^ whereas the concentration of 6.3 ranged from 0.4 to 93 µg mL^−1^ (Appendix A). The radioactivity of plasma samples was measured at 30 min, 2 h, 6 h, and 24 h after injection of ^125^I-PEG-EPO. Both AGP4 and 6.3 dose-dependently accelerated the clearance of PEG-EPO from the blood of mice (Figure 5A,B). Anti-PEG IgM antibody AGP4 at concentrations of 3.4, 11.4, and 57.8 μg mL^−1^ reduced the amount of PEG-EPO remaining in the circulation at 24 h by 54%, 82%, and 91% as compared to mice injected with PBS (sham) at 24 h after PEG-EPO administration (Figure 5C). Anti-PEG IgG antibody 6.3 at a concentration of 0.4 µg mL^−1^ significantly reduced the concentration of PEG-EPO in blood at 24 h by 60% while 6.3 concentrations greater than 3.6 µg mL^−1^ reduced PEG-EPO serum concentrations by 95% at 24 h relative to mice treated with PBS at 24 h after administration of PEG-EPO (Figure 5D), suggesting that the clearance process can be saturated. Control IgM and IgG antibodies at the highest doses did not affect the clearance of PEG-EPO from serum. A molar ratio of AGP4 (Appendix A) or 6.3 (Appendix A) to PEG-EPO of greater than about three appeared to accelerate clearance of 50% PEG-EPO, indicating possible thresholds that result in PEG-EPO clearance.

### 3.6. ^125^I-PEG-EPO Clearance is Mediated by the Fc Portion of Anti-PEG IgG

Clearance induced by anti-PEG 6.3 IgG appeared to be saturable (Figure 5B), suggesting a receptor-mediated mechanism of action. To investigate if clearance of PEG-EPO by 6.3 IgG depends on the Fc portion of the antibody, highly purified 6.3 F(ab’)_2_ fragments were produced (Figure 6A, S4A). 6.3 F(ab’)_2_ fragments displayed similar PEG binding as intact 6.3 IgG (Figure 6A, left) but could not be recognized by an anti-Fc specific secondary antibody (Figure 6A, right), demonstrating the absence of intact IgG in the F(ab’)_2_ preparation. The same amount of 6.3 or 6.3 F(ab’)_2_ were injected into mice 1 h before injection of ^125^I-labeled PEG-EPO. The concentrations of 6.3 and 6.3 F(ab’)_2_ were similar after 1 h when PEG-EPO was injected (Appendix A). In contrast to intact 6.3 IgG, 6.3 F(ab’)_2_ did not accelerate the clearance of PEG-EPO (Figure 6B), indicating that the Fc portion of anti-PEG IgG is important for rapid clearance of PEG-EPO. These results show that pre-existing anti-PEG antibodies can accelerate the clearance of PEG-EPO in mice through Fc-mediated effects.

### 3.7. Fc-Mediated Clearance can be Compensated by Altering PEG-EPO Dose

To examine if the deleterious effects of anti-PEG antibodies on PEG-EPO bioactivity could be overcome, mice were injected with 150 μg kg^−1^ 6.3 antibody 24 h before intravenous injection of increasing doses of PEG-EPO ranging from 5 to 20 μg kg^−1^. Seven days later, the concentration of red blood cells was measured by flow cytometry. 150 μg kg^−1^ 6.3 significantly blocked RBC maturation induced by a normal dose of PEG-EPO (5 μg kg^−1^), but higher doses of PEG-EPO could overcome the effect of 6.3 anti-PEG antibody (Figure 7A). Molar ratios of PEG-EPO to 6.3 exceeding about 1:3 could recapitulate full PEG-EPO activity (Figure 7B). These results show that infusing higher dose of PEG-EPO could be a possible way to overcome the effects of pre-existing anti-PEG antibodies on PEG-EPO.

## 4. Discussion

Methoxy polyethylene glycol-epoetin beta (PEG-EPO) is approved and widely used to treat patients suffering from anemia due to its long circulation time, which can reduce suffering associated with frequent injections. However, about 40% of normal individuals have pre-existing antibodies against PEG which might bind to PEG-EPO and alter its biological activity. We found that pre-existing anti-PEG IgM and IgG antibodies from normal human donors can bind to PEG-EPO. Anti-PEG IgM and IgG monoclonal antibodies altered the biodistribution of PEG-EPO in mice and reduced red blood cell maturation stimulated by PEG-EPO. We also observed a higher incidence of anti-PEG antibodies in anemia patients that failed to respond to PEG-EPO. Taken together, our study suggests that the presence of pre-existing anti-PEG antibodies may alter the therapeutic efficacy of PEG-EPO in anemia patients.

Some pegylated medicines, including pegylated liposomes and some pegylated proteins, can induce high titers of anti-PEG antibodies in animals and humans [6,11,17,18,19,20]. PEG antibodies appear to be induced most often when multiple short PEG chains are attached to foreign proteins (Table 2) [6,7,11]. For example, injections of OVA modified with multiple 5 kDa PEG chains into mice induced anti-PEG IgM antibodies [18]. Pegylated liposomes conjugated with multiple 2 kDa PEG chains can induce anti-PEG IgM antibodies which then accelerate the clearance of a second injection of pegylated liposomes in rats [19]. Induction of anti-PEG antibodies by pegylated proteins derived from microbial or non-human animal sources is also associated with decreased drug efficacy [7,8,11,14]. Armstrong et al. found that patients treated with microbially-derived asparaginase modified with multiple 5 kDa PEG chains developed anti-PEG antibodies that resulted in undetectable asparaginase activity in their blood [7]. Sundy et al. and Lipsky et al. also reported that refractory gout patients treated with porcine uricase (pegloticase) modified with multiple 10 kDa PEG chains developed anti-PEG antibodies that adversely affected drug effectiveness [8,9,10,11].

In contrast to foreign proteins modified with multiple, short PEG molecules, many pegylated biopharmaceutics are derived from human proteins and are conjugated with one long linear or branched PEG chain. [2] PEG-EPO is a good example of such a drug: it is derived from human erythropoietin and is modified with a single linear 30 kDa PEG chain. Until now, there is no evidence that the infusion of PEG-EPO induces the formation of new anti-PEG antibodies in human patients. However, recent reports show that many normal individuals already have anti-PEG antibodies in their circulation, likely due to repeated and long-term use of cosmetics, soaps, and lotions that contain PEG or PEG-like derivatives. The reported prevalence of pre-existing antibodies in humans ranges from about 5% to 70% [8,10,16,28,29,45,46]. This large range can be attributed to different assay sensitivities and the small number of donors examined in some studies. The impact of pre-existing anti-PEG antibodies on the new generation of pegylated drugs is an important open question.

To explore possible effects of pre-existing anti-PEG antibodies on PEG-EPO in vivo, we used a mouse model in which we pre-injected anti-PEG monoclonal antibodies to give serum concentrations covering the range of anti-PEG antibodies measured in human donors, which ranged from 0.1 to 57.3 µg mL^−1^ for anti-PEG IgM antibodies and from 0.3 to 238 µg mL^−1^ for anti-PEG IgG antibodies [25]. This model mimics the presence of pre-existing anti-PEG antibodies during PEG-EPO administration. We injected a dose of ^125^I-labeled PEG-EPO to achieve peak serum concentrations of around 70 ng·mL−1, similar to peak PEG-EPO concentrations in humans based on allometric scaling [39,40,47]. Anti-PEG IgM and IgG antibodies blocked the ability of PEG-EPO to stimulate the formation of new red blood cells in mice. Consistent with our early reports showing that anti-PEG antibodies can accelerate the clearance of a pegylated enzyme in mice [32,33], both IgM and IgG anti-PEG antibodies altered the biodistribution of PEG-EPO dose-dependently with enhanced accumulation in the liver and spleen and a corresponding drop in serum levels. Analysis of the data showed that a 50% reduction in PEG-EPO serum concentration was caused by a molar ratio of about three anti-PEG antibodies to each molecule of PEG-EPO for both anti-PEG IgM and IgG antibodies. We previously reported median concentrations of 0.96 µg mL^−1^ for anti-PEG IgM antibodies and 1.79 µg mL^−1^ for anti-PEG IgG antibodies in 1504 Han Chinese healthy donors [25], implying that anti-PEG antibodies might alter the effectiveness of PEG-EPO in a large number of patients. However, differences in the clearance of immune complexes in mice and humans as well as differences in the affinity of pre-existing anti-PEG antibodies in patients as compared to the monoclonal antibodies used in our study make such estimates difficult. Rather, our results suggest that clinical studies to address the impact of pre-existing anti-PEG antibodies on PEG-EPO and similar pegylated medicines are warranted.

6.3-mediated clearance of PEG-EPO from blood became saturated at doses greater than 3.6 µg mL−1, suggesting specific uptake of immune complexes via a receptor-mediated process. Clearance depended on the Fc portion of 6.3 because 6.3 F(ab’)2 fragments, which bind PEG-EPO equally well as intact 6.3, did not accelerate the clearance of PEG-EPO. Taken together, these results suggest that 6.3-PEG-EPO immune complexes bind and undergo endocytosis via Fc receptors present on resident phagocytes in the spleen and liver. As different subclasses of IgG antibodies bind to Fc receptors with different affinities [48], individuals with similar total anti-PEG levels but with different subclasses of anti-PEG antibodies may experience different alterations in PEG-EPO biological activity.

An important question is whether the effect of pre-existing anti-PEG antibodies can be overcome by increasing the administered dose of PEG-EPO. We found that increasing the dose of PEG-EPO to achieve a molar ratio of PEG-EPO to anti-PEG IgG of about 0.5 or greater could completely compensate for the deleterious effects of anti-PEG antibodies on PEG-EPO biological activity in vivo. In the future, measurement of pre-existing anti-PEG antibodies in patients may help choose the appropriate dose of PEG-EPO. Measurement of pre-existing antibodies may also help increase the safety of PEG-EPO administration. For example, a recent case report described anaphylaxis reactions in a patient immediately after the first administration of PEG-EPO even though the patient previously received epoetin-α without complications, [49] suggesting that the presence of pre-existing anti-PEG antibodies may have been responsible for the adverse effects to PEG-EPO. Our report shows that anti-PEG antibodies can alter the activity of PEG-EPO with one linear 30 kDa chain and suggests that additional studies are warranted to delineate the effect of pre-existing anti-PEG antibodies because pegylated drugs with one long PEG chain are being vigorously developed in clinical trials.

## Figures and Tables

**Figure 1 pharmaceutics-12-00015-f001:**
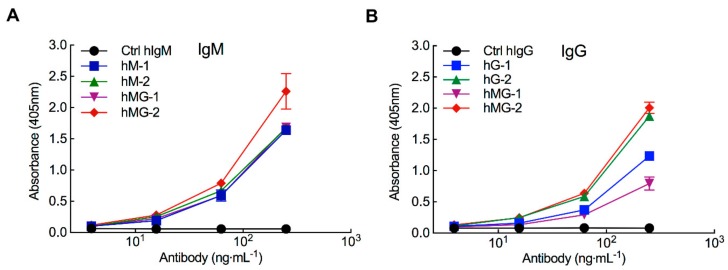
Pre-existing human anti-PEG antibodies bind to methoxy polyethylene glycol-epoetin beta (PEG-EPO). Serial dilutions of plasma samples from donors that were previously determined to be positive for anti-PEG IgM (hM1, hM2), anti-PEG IgG (hG1, hG2), or both anti-PEG IgM and IgG (hMG1 and hMG2) were assayed for anti-PEG IgM (**A**) or anti-PEG IgG (**B**) binding to PEG-EPO in 96-well plates using anti-human IgM or IgG specific secondary antibodies. Control IgM (Ctrl hIgM) and control IgG (Ctrl hIgG) are negative control human IgM and IgG antibodies, respectively (bars, SD; *n* = 2).

**Figure 2 pharmaceutics-12-00015-f002:**
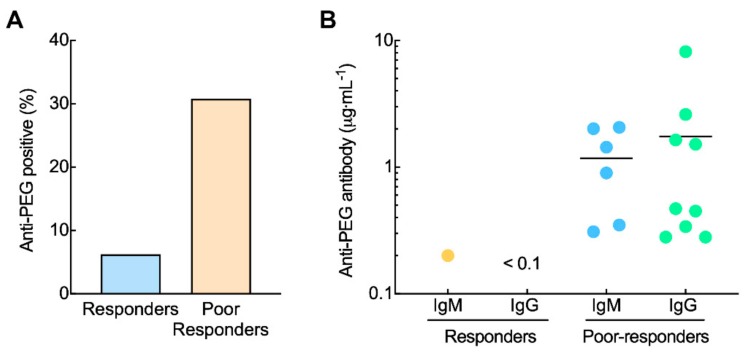
Anti-PEG antibodies may alter the therapeutic efficacy of PEG-EPO. Blood samples were collected from 39 peritoneal dialysis patients who did not respond well to PEG-EPO and 16 control patients who did respond to PEG-EPO. The hematocrit and concentrations of pre-existing anti-PEG antibodies were measured. The prevalence and concentrations of positive pre-existing anti-PEG antibodies within responders and poor-responders are shown in (**A**,**B**).

**Figure 3 pharmaceutics-12-00015-f003:**
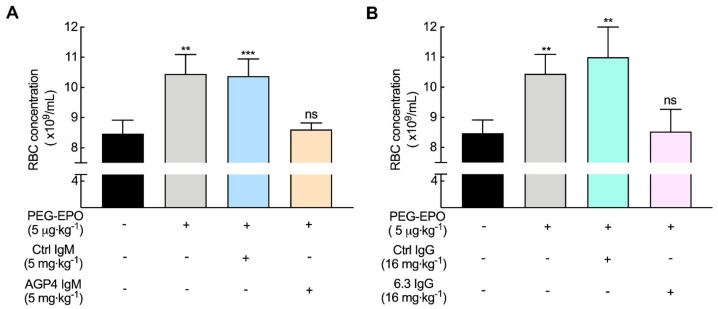
Pre-existing anti-PEG antibodies can alter the bioactivity of PEG-EPO in mice. (**A**,**B**) Female BALB/c mice were pre-injected with 5 mg kg^−1^ AGP4 or 16 mg kg^−1^ 6.3 anti-PEG antibodies at 24 h before intravenously injection of 5 μg kg^−1^ PEG-EPO. RBC numbers were measured on day 7. Control IgM (Ctrl IgM) and control IgG (Ctrl IgG) are non-binding control mouse IgM and IgG antibodies, respectively (*n* = 4). Significant differences between mean RBC numbers in mice receiving antibodies versus sham-treated mice are indicated: ns, not significant; ^**^, *p* < 0.01; ^***^, *p* < 0.005.

**Figure 4 pharmaceutics-12-00015-f004:**
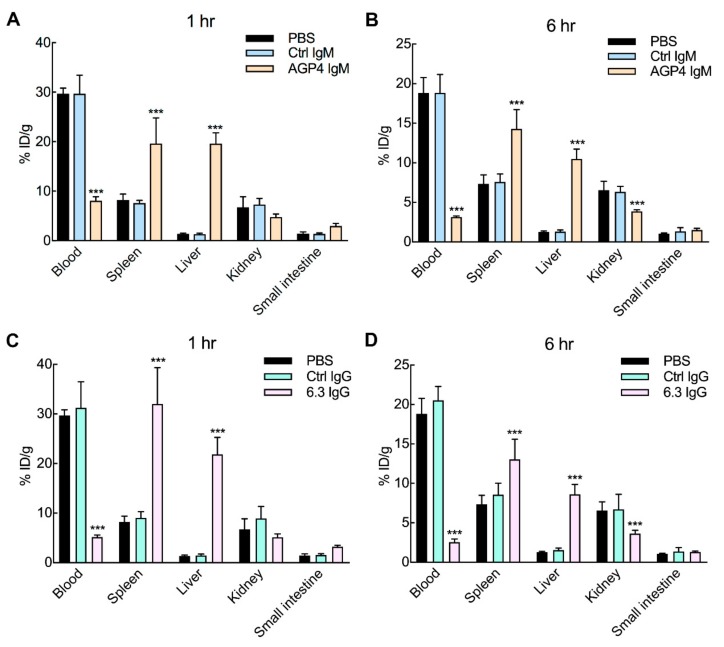
PEG-EPO cleared by pre-existing anti-PEG antibodies accumulates in the spleen and liver. Female BALB/c mice were injected with 5 mg kg^−1^ AGP4 or 16 mg kg^−1^ 6.3 IgG anti-PEG antibodies 24 h before intravenous infusion of 5 μg kg^−1 125^I-PEG-EPO. Mice were sacrificed and perfused at 1 h (**A**,**C**) and 6 h (**B**,**D**) after PEG-EPO injection. Results show mean values of radioactivity in plasma and organs as percent injected dose per gram of tissue. Control IgM (Ctrl IgM) and control IgG (Ctrl IgG) are non-binding mouse IgM and IgG antibodies, respectively (*n* = 4). Significant differences between mean values of PBS-treated (sham) and other mice are indicated: ^***^
*p* < 0.005.

**Figure 5 pharmaceutics-12-00015-f005:**
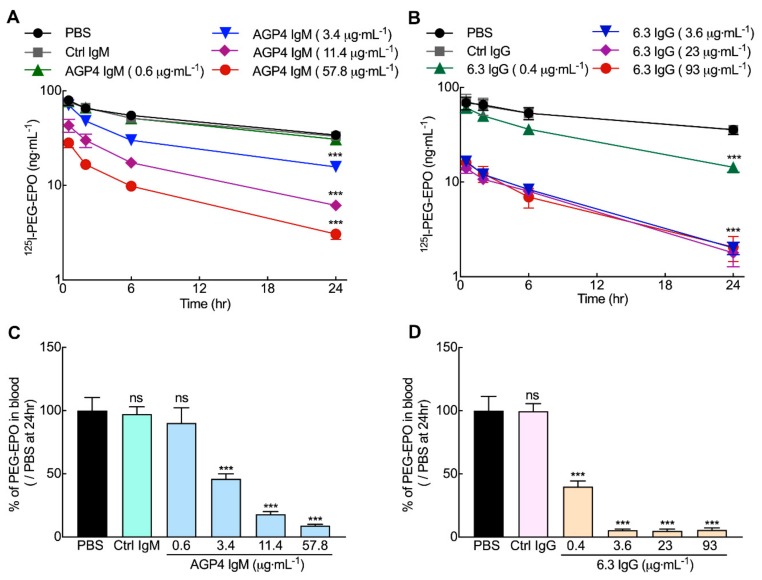
Anti-PEG antibodies accelerate clearance of PEG-EPO in mice. Female BALB/c mice were injected with different doses of mouse anti-PEG AGP4 IgM (**A**) or 6.3 IgG (**B**) antibodies 24 h before intravenous injection of 5 μg kg^−1 125^I-PEG-EPO. Radioactivity was measured in plasma samples collected at 30 min, 2 h, 6 h, and 24 h. The percentage of PEG-EPO remaining in the circulation at different doses of AGP4 (**C**) or 6.3 (**D**) at 24 h as compared to PEG-EPO injection only are shown. Control IgM (Ctrl IgM) and control IgG (Ctrl IgG) are negative control mouse antibodies that could not bind PEG-EPO, respectively (*n* = 4). Significant differences in mean values in comparison to mice treated with PEG-EPO alone are indicated: ns, not significant; ^***^, *p* < 0.005.

**Figure 6 pharmaceutics-12-00015-f006:**
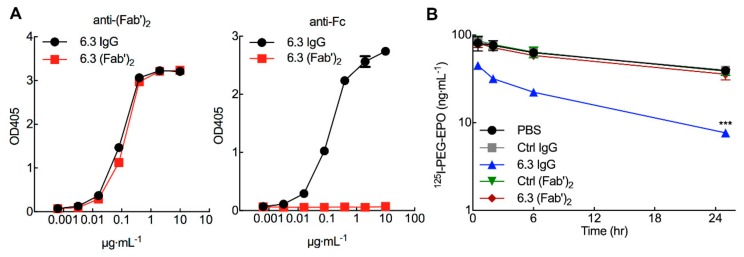
Anti-PEG mediates clearance of PEG-EPO via the Fc portion of IgG. (**A**) Binding of 6.3 IgG and 6.3 F(ab’)_2_ was measured by direct ELISA on plates coated with 0.5 μg/well NH_2_-PEG_10,000_-NH_2_ and detected by peroxidase-conjugated goat anti-mouse F(ab’)_2_ (left) or peroxidase-conjugated goat anti-mouse Fc antibodies (right). (**B**) 200 μg kg^−1^ control IgG, 6.3 IgG, control F(ab’)_2_ or 6.3 F(ab’)_2_ were intravenously injected into mice (*n* = 4) 1 h before injection of 5 μg kg^−1 125^I-PEG-EPO. Results show the concentration of ^125^I-PEG-EPO in serum samples. Significant differences in mean values relative to mice treated with PBS are indicated: ^***^, *p* < 0.005.

**Figure 7 pharmaceutics-12-00015-f007:**
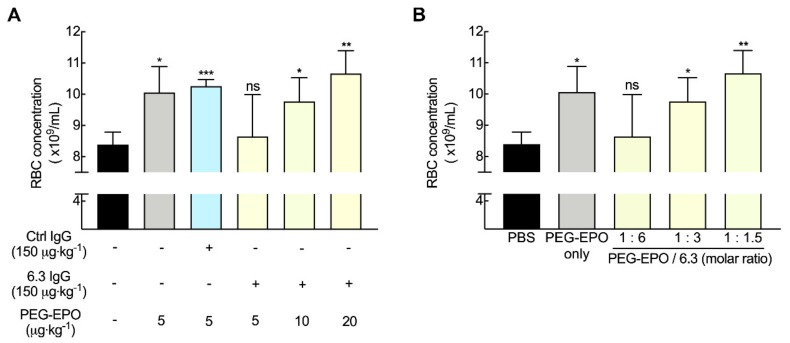
Inhibitory effect of anti-PEG can be compensated by higher PEG-EPO dose. (**A**) Different doses of PEG-EPO (5, 10 and 20 μg kg^−1^) were intravenously injected at 24 h after injection of 150 μg kg^−1^ 6.3 IgG (*n* = 4). The concentrations of RBCs on day 7 are shown. Significant differences in comparison to mice treated with PBS are indicated: ns, not significant; ^*^, *p* < 0.05; ^**^, *p* < 0.01; ^***^, *p* < 0.005. (**B**) The RBC numbers at different molar ratios of PEG-EPO to 6.3 IgG are shown (*n* = 4). Significant differences in comparison to mice treated with PBS are indicated: ns, not significant; ^*^, *p* < 0.05; ^**^, *p* < 0.01.

**Table 1 pharmaceutics-12-00015-t001:** FDA-approved pegylated drugs.

Brand Name	Component	PEG (kDa)	PEG Number	Disease	Approval Year
Adagen^®^	Adenosine deaminase	5	11-17	Severe immunodeficiency	1990
Oncaspar^®^	L-Asparaginase	5	69-82	Leukemia	1994
Doxil^®^	Liposomal doxorubicin	2	multiple	Cancer	1995
PEG-Intron^®^	Interferon alfa-2b	12	1	Hepatitis C	2001
PEGASYS^®^	Interferon alfa-2a	40	1 (branched)	Hepatitis	2001
Neulasta^®^	G-CSF	20	1	Neutropenia	2002
Somavert^®^	Antagonist (GHR)	5	4-6	Acromegaly	2003
Macugen^®^	Anti-VEGF aptamer	20	2	Macular degeneration	2004
Mircera^®^	Epoetin beta	30	1	Anemia	2007
Cimzia^®^	Anti-TNFα Fab	40	1 (branched)	Rheumatoid arthritis	2008
Krystexxa^®^	Uricase	10	9-11	Gout	2010
Sylatron^®^	Interferon alfa-2b	12	1	Melanoma	2011
Omontys^®^	Analog of erythropoietin	40	1 (branched)	Anemia	2012
Movantik^®^	Antagonist (C_34_H_53_NO_11_)	0.3	1	Opioid-induced constipation	2014
Plegridy^®^	Interferon beta-1a	20	1	Multiple sclerosis	2014
Adynovate^®^	Factor VIII	20	1 (branched)	Hemophilia A	2015
Onivyde^®^	Liposomal irinotecan	2	multiple	Cancer	2015
Rebinyn^®^	Factor IX	40	1	Hemophilia B	2017
Palynziq^®^	Phenylalanine ammonia lyase	20	9	Phenylketonuria	2018
Jivi^®^	Factor VIII	60	1 (branched)	Hemophilia A	2018

**Table 2 pharmaceutics-12-00015-t002:** Accelerated clearance of pegylated agents by anti-polyethylene glycol (PEG) antibodies.

Component	PEG (kDa)	PEG Number	Reference
β-glucuronidase	5	multiple	[33,34]
β-glucuronidase	2, 5, 20	multiple	[35]
OVA	40	multiple	[18]
Liposomes	2	multiple	[19,20,21,22,23,36,37]

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
