# Peer review of "Both IgM and IgG Antibodies against Polyethylene Glycol Can Alter the Biological Activity of Methoxy Polyethylene Glycol-Epoetin Beta in Mice"

_pharmaceutics, 2019, doi:10.3390/pharmaceutics12010015_

Round 1

Reviewer 1 Report

This is a well-written and significant contribution to the field of PEG-modified biologics. The authors develop relevant mouse models, and demonstrate that serum titers of anti-PEG IgG or IgM antibodies significantly attenuate the activities of mono-PEG containing EPO. I only have a few minor points.

In the introduction, the authors state that >40% of people have measurable anti-PEG antibodies. If the existence of such antibodies is truly deleterious to treatment with PEG-biologics, why did only 30.8% of patients with pre-existing titers respond poorly to PEG-EPO? Was it because the titers were very low, or perhaps factors not having anything to do with anti-PEG antibodies are responsible? Can the effects of the AGP4 and 6.3 antibodies be overcome using IvIg? Would IvIg be advisable as a co-administered blocker with PEG-EPO? The Fab'2 experiment is interesting, but no PK is provided. The lack of efficacy may simply be due to accelerated clearance, and not to FcR binding. This could be a discussion item, or a qualification point. For Fig. 7, what would the RBC levels be with 10 and 20 ug/kg PEG-EPO? No additional experiments are needed, but that information would be good to know. Very small point: the authors could use another round of minor editing for such things as 24h, 24 hours, twenty four hours, and 24 hour. Sorry to point that out, but I'd do it if I were them.

Author Response

Thank you.

Reviewer 2 Report

This manuscript by Chang et al studies the impact of pre-existing anti-PEG antibodies on the clearance and activity of PEGylated protein therapeutic PEG-EPO in mice.  This study builds on the teams' previous work on this topic. 

There are many published articles on accelerated blood clearance (ABC) of PEGylated agents.  This adds some additional insight by specifically (1) measuring pre-existing anti-PEG antibodies in humans, and (2) testing the impact of those physiological levels of anti-PEG antibodies in mouse models of ABC.  The studies appear appropriately designed and executed.   The manuscript is relatively clear and well written.   However, the primary question relates to the doses used in mice, as indicated below.

Comments:

In Results section 3.1, the identity of the human pre-existing antibodies (hM1, hM2, hG1, hG2, hMG1, hMG2) is not clear.   What were the anti-PEG antibody titers in the respective patients from which these were obtained?   Were they from high/medium/low titer patients?  This is also not clearly defined in their previous Chen et al (2016 Analytical Chemistry) paper.  The range of pre-existing antibodies listed is very broad (0.1 - 57.3 ug/ml IgM; 0.3 - 238 IgG).  In this section, there is value to indicate the mean and median +/- (standard dev or std error?) values that are listed in the Chen et al 2016 paper.   With a mean/median antibody levels from ~1 - 5 ug/ml, these are within range of the ~1.2 - 1.7 ug/ml correlated with poor responders in Fig 2, so that is OK.  But these values are much lower than the 58 ug/ml IgM and 93 ug/ml IgG  used in the mouse model (section 3.3).   These were achieved by dosing 5 and 16 mg/kg anti-PEG antibody.  Why were those target levels selected and how do they translate to humans?   It would be good to know the impact of the lower concentration range on activity in these models.

Minor comments:

Table 1:   A list of approved drugs is not typical in a non-review, but this is a helpful resource to readers so it is worth retaining.

Author Response

Thank you.
